# Permanent pacemaker implantation after On-X surgical aortic valve replacement: SWEDEHEART observational study

Magnus Dalén ,[1,2] Michael Persson,[1,2] Natalie Glaser ,[2,3] Ulrik Sartipy [1,2]

¹Department of Cardiothoracic Surgery, Karolinska University Hospital, Stockholm, Sweden
²Department of Molecular Medicine and Surgery, Karolinska Institutet, Stockholm, Sweden
³Department of Cardiology, Stockholm South General Hospital, Stockholm, Sweden

**Correspondence to**
Dr Magnus Dalén;
magnus.dalen@ki.se

## ABSTRACT

**Objective** Bioprosthetic aortic valves with an extended subannular component, such as transcatheter valves, exert increased compression on the cardiac conduction system and increase the risk for permanent pacemaker implantation. It is unknown if the On-X mechanical prosthetic valve, which has an elongated subannular valve housing, increases the risk of permanent pacemaker implantation following aortic valve replacement.
**Design** Observational nationwide cohort study.
**Setting** Swedish population-based study.
**Participants** All patients aged 18–65 years who underwent primary mechanical aortic valve replacement in Sweden between 2005 and 2018. We used the Swedish Web system for Enhancement and Development of Evidence-based care in Heart disease Evaluated According to Recommended Therapies register and other Swedish national health-data registers.
**Exposure** Patients implanted with an On-X valve versus patients implanted with other bileaflet mechanical valves.
**Primary and secondary outcome measures** Primary outcome measure was permanent pacemaker implantation within 30 days of surgery.
**Results** A total of 2602 patients were included, and 581 patients received an On-X valve and 2021 patients received a St Jude Masters/Regent (n=945) or Carbomedics Reduced valve (n=1076). In the total study population, 115 (4.4%) permanent pacemaker implantations were performed within 30 days after aortic valve replacement. In the propensity score matched population, there was no significant difference in the rate of permanent pacemaker implantation in the On-X group compared with the control group: 3.6% (95% CI: 2.4% to 5.5%) vs 4.0% (95% CI: 2.7% to 5.9%), p=0.877.
**Conclusions** The On-X prosthetic heart valve was associated with a similarly low risk for permanent pacemaker implantation after aortic valve replacement compared with other conventional bileaflet mechanical valves. The On-X elongated subannular valve housing does not interfere with the cardiac conduction system.

## Strengths and limitations of this study

► Data were obtained from high-quality national Swedish health data registers.
► Long and complete follow-up in a nationwide cohort.
► One limitation of the study was that data did not include preoperative conduction disorders known to increase the risk for postoperative pacemaker requirement, such as bundle branch block.
► Another limitation was that we had no information about specific indications of postoperative pacemaker implantation.

## INTRODUCTION

Surgical aortic valve replacement can induce cardiac conduction disorders that require permanent pacemaker implantation in 2%–6% of the patients.[1–4] The close proximity of the atrioventricular node and the bundle of His to the aortic valve annulus and the left ventricular outflow tract makes the cardiac conduction system prone to injury during implantation of an aortic valve prosthesis.[3] Compared with conventional intra-/supraannulary prosthetic valves, bioprosthetic valves with an extended subannular component, such as transcatheter and rapid deployment valves, exerts increased compression on the conduction system.[5 6] These prosthetic valves, as well as the subannular implantation depth, are associated with markedly increased risks for permanent pacemaker implantation.[5–8] The current standard for mechanical aortic valves used in contemporary clinical practice are bileaflet valves implanted intra- or supraannulary. The On-X prosthetic valve (On-X Life Technologies, Austin, Texas) is one of the most commonly used bileaflet mechanical valves today. Its' design differs from other contemporary mechanical valves,[9] and it is the only mechanical heart valve approved for reduced international normalised ratio target range.[10 11] In order to reduce turbulence and enhance blood flow over the prosthetic valve, the On-X consists of an elongated valve housing with a flared inlet that protrudes subannulary into the left ventricular outflow tract.[9] It is unknown

whether this design increases the risk for conduction disorders requiring permanent pacemaker implantation.

We performed a nationwide population-based cohort study to analyse whether aortic valve replacement with implantation of the On-X prosthetic heart valve, compared with other conventional bileaflet mechanical valves, was associated with an increased risk for permanent pacemaker implantation.

## METHODS

This nationwide population-based observational cohort study followed the Strengthening the Reporting of Observational Studies in Epidemiology and the REporting of studies Conducted using Observational Routinely collected health Data guidelines for observational studies using routinely collected data.[12 13] The study was approved by the Swedish Ethical Review Authority and the need for informed consent was waived (Registration number: 2019–04131).

### Study population

The Swedish Web system for Enhancement and Development of Evidence-based care in Heart disease Evaluated According to Recommended Therapies register[14 15] was used to identify all adult patients aged 18–65 who underwent primary mechanical aortic valve replacement between 2005 and 2018. Patients who had a permanent pacemaker or implantable cardioverter-defibrillator prior to surgery or who underwent valve replacement because of infective endocarditis, or who had concomitant mitral valve surgery were excluded. Implantation of the On-X prosthetic valve was the exposure and the two most common valve types (Carbomedics Reduced (LivaNova, London, United Kingdom) and St Jude Masters/Regent (St Jude Medical, St Paul, Minnesota, USA)) served as the control group. Thus, patients who received other valve models were excluded.

As previously described, individual-level data linking to other nationwide healthcare registries was performed using the unique personal identity numbers assigned to all Swedish residents.[16 17] The National Patient Register[18] was used to acquire information regarding relevant prior medical history, and the LISA (Longitudinal integration database for health insurance and labour market studies) database, managed by Statistics Sweden, was used to obtain information regarding socioeconomic variabels.[19]

### Outcomes

The primary outcome measure was the implantation of a permanent pacemaker or implantable cardioverter-defibrillator within 30 days following aortic valve replacement as identified by the International Classification of Diseases-codes (FPE00, FPE10, FPE20, FPE26, FPF00, FPF10, FPF20, FPG10, FPG20, FPG30 and FPG33) from the Swedish National Patient Register.[18] We included implantable cardioverter defibrillator in the primary outcome measure in order to capture patients who had dual indications for both pacing and defibrillation.

### Statistical methods

Baseline characteristics were described with frequencies and percentages for categorical variables and means and SD for continuous variables. To minimise confounding due to measured baseline covariates, we estimated propensity scores (the probability of receiving an On-X valve based on the observed data) using a logistic regression model that included all variables reported in table 1. A propensity score matched cohort was created by 1:1 nearest neighbour matching on the logit of the propensity score without replacement and a calliper width of 0.2 times the SD of the logit of the propensity score.[20] In a separate analysis, we also estimated propensity scores using generalised boosted regression modeling[21 22] and used the scores for inverse probability of treatment weighting. Finally, we applied the propensity scores from a logistic regression model to construct overlap weights, a weighting method that yields exact covariate balance between the exposed and reference groups by construction.[23] Balance between the groups was assessed by standardised mean differences. An absolute standardised difference ≤0.1 was considered an ideal balance.[24] In the matched and weighed populations, respectively, the percentage of pacemaker implants and 95% CI were compared between the groups using McNemar's and $\chi^2$ tests, respectively. To illustrate the pacemaker implantation rate the first 90 days postoperatively, we constructed Kaplan-Meier curves. Although death could be considered a competing event, it occurred rarely during the first 90 days of follow-up and was therefore not accounted for. The statistical analyses were performed with R version 4.0.2 (R Foundation for Statistical Computing, Vienna, Austria) and the Matching[25] and twang[22] packages.

### Missing data

Although data were complete for most variables, including exposure and outcome, there were some missing data. The variables with missing data were body mass index, estimated glomerular filtration rate, valve size and educational level with 6.9%, 1.5%, 0.7% and 0.6% missing data, respectively. For these variables, missing data was imputed with the most common category. Missing data was handled differently in the inverse probability of treatment weighted analysis because the weights were constructed to also balance rates of missingness in both groups.[21 22]

### Patient and public involvement

There was no patient or public involvement in the design or conduct of this study.

## RESULTS

A total of 2602 patients were included in the study, and 581 patients received an On-X valve and 2021 patients

**Table 1** Baseline characteristics in patients who underwent aortic valve replacement in Sweden with an On-X valve or other mechanical valves before and after propensity score matching

| | Before matching | | | | After propensity score matching | | | |
|---|---|---|---|---|---|---|---|---|
| | Overall | Control | On-X | SMD | Overall | Control | On-X | SMD |
| Number of patients | 2602 | 2021 | 581 | | 1160 | 580 | 580 | |
| Age (years), mean (SD) | 53.4 (9.5) | 54.1 (9.0) | 51.0 (10.7) | 0.322 | 51.4 (10.6) | 51.9 (10.6) | 51.0 (10.6) | 0.079 |
| Female sex | 631 (24.3) | 525 (26.0) | 106 (18.2) | 0.187 | 207 (17.8) | 101 (17.4) | 106 (18.3) | 0.023 |
| Non-Nordic birth region | 239 (9.2) | 207 (10.2) | 32 (5.5) | 0.176 | 52 (4.5) | 20 (3.4) | 32 (5.5) | 0.100 |
| Educational level | | | | 0.147 | | | | 0.017 |
| <10 years | 720 (27.7) | 588 (29.1) | 132 (22.7) | | 266 (22.9) | 135 (23.3) | 131 (22.6) | |
| 10–12 years | 1263 (48.5) | 959 (47.5) | 304 (52.3) | | 604 (52.1) | 300 (51.7) | 304 (52.4) | |
| >12 years | 619 (23.8) | 474 (23.5) | 145 (25.0) | | 290 (25.0) | 145 (25.0) | 145 (25.0) | |
| Disposable household income (quartiles) | | | | 0.119 | | | | 0.074 |
| Q1 (low) | 651 (25.0) | 520 (25.7) | 131 (22.5) | | 260 (22.4) | 129 (22.2) | 131 (22.6) | |
| Q2 | 651 (25.0) | 506 (25.0) | 145 (25.0) | | 289 (24.9) | 144 (24.8) | 145 (25.0) | |
| Q3 | 650 (25.0) | 512 (25.3) | 138 (23.8) | | 262 (22.6) | 124 (21.4) | 138 (23.8) | |
| Q4 (high) | 650 (25.0) | 483 (23.9) | 167 (28.7) | | 349 (30.1) | 183 (31.6) | 166 (28.6) | |
| Married | 1326 (51.0) | 1065 (52.7) | 261 (44.9) | 0.156 | 531 (45.8) | 270 (46.6) | 261 (45.0) | 0.031 |
| Body mass index (tertiles) | | | | 0.053 | | | | 0.082 |
| T1 (low) | 808 (31.1) | 623 (30.8) | 185 (31.8) | | 387 (33.4) | 203 (35.0) | 184 (31.7) | |
| T2 | 987 (37.9) | 778 (38.5) | 209 (36.0) | | 418 (36.0) | 209 (36.0) | 209 (36.0) | |
| T3 (high) | 807 (31.0) | 620 (30.7) | 187 (32.2) | | 355 (30.6) | 168 (29.0) | 187 (32.2) | |
| Atrial fibrillation | 233 (9.0) | 183 (9.1) | 50 (8.6) | 0.016 | 99 (8.5) | 49 (8.4) | 50 (8.6) | 0.006 |
| Heart failure | 382 (14.7) | 310 (15.3) | 72 (12.4) | 0.085 | 134 (11.6) | 62 (10.7) | 72 (12.4) | 0.054 |
| Left ventricular ejection fraction (%) | | | | 0.079 | | | | 0.015 |
| >50 | 1953 (75.1) | 1518 (75.1) | 435 (74.9) | | 872 (75.2) | 437 (75.3) | 435 (75.0) | |
| 30–50 | 511 (19.6) | 389 (19.2) | 122 (21.0) | | 239 (20.6) | 118 (20.3) | 121 (20.9) | |
| <30 | 138 (5.3) | 114 (5.6) | 24 (4.1) | | 49 (4.2) | 25 (4.3) | 24 (4.1) | |
| Chronic obstructive pulmonary disease | 155 (6.0) | 136 (6.7) | 19 (3.3) | 0.159 | 33 (2.8) | 14 (2.4) | 19 (3.3) | 0.052 |
| Diabetes | 361 (13.9) | 301 (14.9) | 60 (10.3) | 0.138 | 114 (9.8) | 54 (9.3) | 60 (10.3) | 0.035 |
| eGFR (ml/min/1,73 m2) | | | | 0.153 | | | | 0.068 |
| >60 | 2457 (94.4) | 1894 (93.7) | 563 (96.9) | | 1124 (96.9) | 562 (96.9) | 562 (96.9) | |
| 45–59 | 97 (3.7) | 85 (4.2) | 12 (2.1) | | 22 (1.9) | 10 (1.7) | 12 (2.1) | |
| 30–44 | 28 (1.1) | 24 (1.2) | 4 (0.7) | | 11 (0.9) | 7 (1.2) | 4 (0.7) | |
| <30 | 20 (0.8) | 18 (0.9) | 2 (0.3) | | 3 (0.3) | 1 (0.2) | 2 (0.3) | |
| Preoperative dialysis | 21 (0.8) | 19 (0.9) | 2 (0.3) | 0.075 | 2 (0.2) | 0 (0.0) | 2 (0.3) | 0.083 |
| Prior myocardial infarction | 235 (9.0) | 193 (9.5) | 42 (7.2) | 0.084 | 91 (7.8) | 49 (8.4) | 42 (7.2) | 0.045 |
| Prior percutaneous coronary intervention | 127 (4.9) | 102 (5.0) | 25 (4.3) | 0.035 | 55 (4.7) | 30 (5.2) | 25 (4.3) | 0.041 |
| Peripheral vascular disease | 305 (11.7) | 208 (10.3) | 97 (16.7) | 0.188 | 184 (15.9) | 87 (15.0) | 97 (16.7) | 0.047 |
| Hypertension | 920 (35.4) | 703 (34.8) | 217 (37.3) | 0.053 | 431 (37.2) | 215 (37.1) | 216 (37.2) | 0.004 |
| Hyperlipidemia | 433 (16.6) | 344 (17.0) | 89 (15.3) | 0.046 | 187 (16.1) | 98 (16.9) | 89 (15.3) | 0.042 |
| Prior stroke | 136 (5.2) | 105 (5.2) | 31 (5.3) | 0.006 | 63 (5.4) | 32 (5.5) | 31 (5.3) | 0.008 |
| History of cancer | 128 (4.9) | 98 (4.8) | 30 (5.2) | 0.014 | 69 (5.9) | 39 (6.7) | 30 (5.2) | 0.066 |

Continued

**Table 1** Continued

| | Before matching | | | | After propensity score matching | | | |
|---|---|---|---|---|---|---|---|---|
| | Overall | Control | On-X | SMD | Overall | Control | On-X | SMD |
| Alcohol dependence | 84 (3.2) | 69 (3.4) | 15 (2.6) | 0.049 | 28 (2.4) | 13 (2.2) | 15 (2.6) | 0.022 |
| Liver disease | 21 (0.8) | 18 (0.9) | 3 (0.5) | 0.045 | 7 (0.6) | 4 (0.7) | 3 (0.5) | 0.022 |
| Prior bleeding event | 82 (3.2) | 67 (3.3) | 15 (2.6) | 0.043 | 27 (2.3) | 12 (2.1) | 15 (2.6) | 0.034 |
| Prior endocarditis | 118 (4.5) | 92 (4.6) | 26 (4.5) | 0.004 | 49 (4.2) | 23 (4.0) | 26 (4.5) | 0.026 |
| Emergent operation | 43 (1.7) | 32 (1.6) | 11 (1.9) | 0.024 | 20 (1.7) | 9 (1.6) | 11 (1.9) | 0.026 |
| Coronary artery bypass grafting | 442 (17.0) | 363 (18.0) | 79 (13.6) | 0.120 | 164 (14.1) | 85 (14.7) | 79 (13.6) | 0.030 |
| Valve size (mm) | | | | 0.390 | | | | 0.030 |
| 19 | 106 (4.1) | 74 (3.7) | 32 (5.5) | | 63 (5.4) | 31 (5.3) | 32 (5.5) | |
| 21 | 493 (18.9) | 376 (18.6) | 117 (20.1) | | 238 (20.5) | 121 (20.9) | 117 (20.2) | |
| 23 | 998 (38.4) | 739 (36.6) | 259 (44.6) | | 520 (44.8) | 262 (45.2) | 258 (44.5) | |
| 25 | 741 (28.5) | 584 (28.9) | 157 (27.0) | | 308 (26.6) | 151 (26.0) | 157 (27.1) | |
| 27 | 264 (10.1) | 248 (12.3) | 16 (2.8) | | 31 (2.7) | 15 (2.6) | 16 (2.8) | |

Numbers are n (%) unless otherwise specified.
eGFR, estimated glomerular filtration rate; SMD, standardized mean difference.

received a St Jude Masters/Regent (n=945) or Carbomedics Reduced valve (n=1076) at the eight hospitals performing cardiac surgery in Sweden during the study period. The proportion of On-X valve implantations increased during the study period but varied markedly by hospital as shown in online supplemental figures 1 and 2 and online supplemental tables 1 and 2. The mean age in the total study population was 53.4 years, and 24% were women. Before propensity score matching, there were differences in baseline characteristics between the groups as shown in table 1. There were more women and the mean age was higher in the control group. Peripheral vascular disease was more common in the On-X valve group, whereas pulmonary disease, diabetes and concomitant coronary artery bypass grafting were more common in the control group. The 30-day all-cause mortality was 0.69% in the On-X valve group vs 0.64% in the control group (p=0.905). After propensity score matching, the two groups were well balanced across all baseline characteristics and no standardised mean differences were greater than 10% (table 1 and online supplemental figures 3 and 4).

### Permanent pacemaker implantation following mechanical aortic valve replacement

In the total study population, 115 (4.4%) permanent pacemaker implantations were performed within 30 days after aortic valve replacement. In the On-X group, the number of pacemakers implanted was 21 (3.6%) vs 94 (4.6%) in the control group (p=0.284). Implantable cardioverter-defibrillators accounted for 5% of the implantations in both groups. The pacemaker implantation rate increased slightly during the study period but was fairly similar between hospitals (online supplemental figures 5 and 6 and online supplemental tables 3 and 4).

The Kaplan-Meier estimated rate of permanent pacemaker implantation in the propensity score matched population is shown in figure 1. There was no significant difference in the rate of permanent pacemaker implantation within 30 days after aortic valve replacement with the On-X valve compared with the control group: 3.6% (95% CI: 2.4% to 5.5%) vs 4.0% (95% CI: 2.7% to 5.9%), p=0.877. The OR (95% CI) for a permanent pacemaker implantation within 30 days after On-X aortic valve replacement was 0.91 (0.49–1.67). The majority of

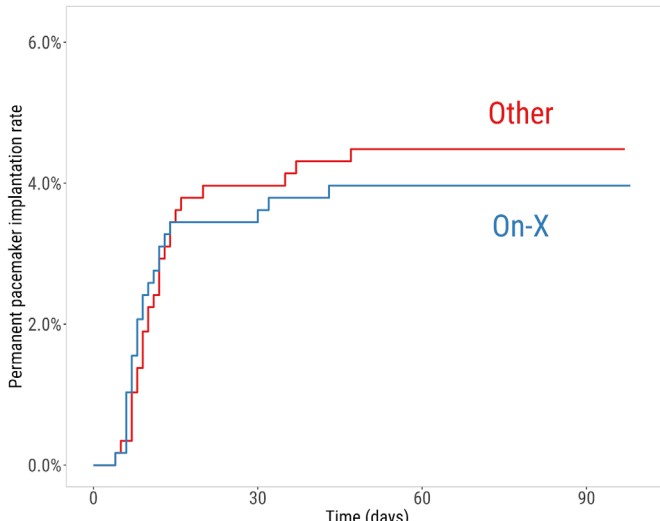

**Figure 1** The graph shows the Kaplan-Meier estimated permanent pacemaker implantation rate in propensity score matched patients who received an On-X valve compared with other valves. There was no significant difference in the pacemaker implant rate between the groups. The majority of permanent pacemaker implantations occurred during the first 30 days. No pacemakers were implanted from day 60 to day 120.

permanent pacemaker implantations occurred during the first 30 days. No pacemakers were implanted from postoperative day 60 to postoperative day 120.

## Alternative approaches for confounding adjustment

Two alternative approaches for confounding adjustment using propensity scores for construction of weights were conducted: inverse probability of treatment weighting and overlap weighting. Both approaches resulted in excellent balance in baseline characteristics between the groups (online supplemental tables 5 and 6 and online supplemental figures 7 and 8). The results were very similar to the results obtained in the propensity score matched analysis, and we found no significant difference in the rate of permanent pacemaker implantation within 30 days after aortic valve replacement with an On-X valve compared with the control group.

In the inverse probability of treatment weighted population, the rate of permanent pacemaker implantation within 30 days after aortic valve replacement was 3.8% (95% CI: 2.0% to 5.6%) with an On-X valve vs 4.7% (95% CI: 3.7% to 5.6%) in the control group, p=0.428. The odds ratio (95% CI) for a permanent pacemaker implantation within 30 days after On-X aortic valve replacement was 0.81 (0.48–1.37).

In the population where overlap weights were used, the rate of permanent pacemaker implantation within 30 days after aortic valve replacement was 3.7% (95% CI: 2.1% to 5.3%) with an On-X valve vs 4.7% (95% CI: 3.6% to 5.8%) in the control group, p=0.357. The odds ratio (95% CI) for a permanent pacemaker implantation within 30 days after On-X aortic valve replacement was 0.79 (0.47–1.31).

The Kaplan-Meier estimated rates of permanent pacemaker implantation in the inverse probability of treatment weighted population, and the population where overlap weights were used, respectively, are shown in online supplemental figures 9 and 10.

The results from all approaches are summarised in online supplemental figure 11. As height and weight were not included in the propensity score model, these data are presented in online supplemental table 7.

## DISCUSSION

The On-X prosthetic heart valve has an elongated subannular valve housing with a flared inlet that protrudes subannulary into the left ventricular outflow tract. This design distinguishes the On-X prosthetic valve from other contemporary bileaflet mechanical aortic valves that are implanted intra-/supraannulary and that do not extend subannulary (figure 2). In this nationwide cohort study, aortic valve replacement with implantation of the On-X prosthetic heart valve was not associated with an increased risk for permanent pacemaker implantation compared with other intra-/supraannulary bileaflet mechanical valves.

Cardiac conduction abnormalities requiring permanent pacemaker implantation is a well-known complication of aortic valve replacement.[1] The widespread use of transcatheter and rapid deployment prosthetic valves, which are associated with a significant risk of injury to the conduction system, has generated an increased interest in research concerning permanent pacemaker implantation after aortic valve replacement.[5–7] Although the use of mechanical prosthetic valves has decreased during later years, these valves are still the primary choice for younger patients, for whom mechanical valves have been shown to be superior to biological valves, in terms of mortality and reoperation rates.[17 26] Permanent pacemaker requirement has in some studies been associated with adverse events and impaired long-term survival.[2]

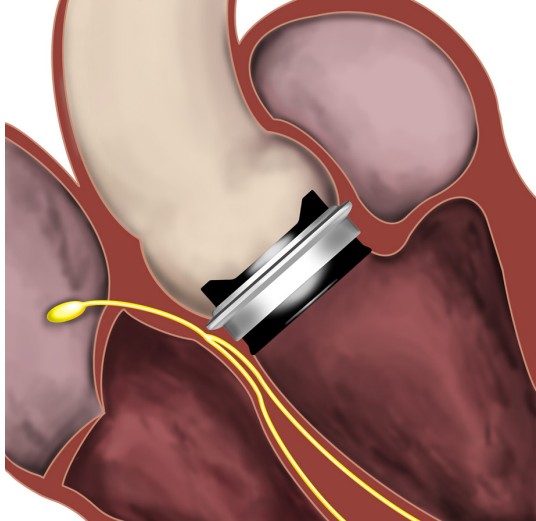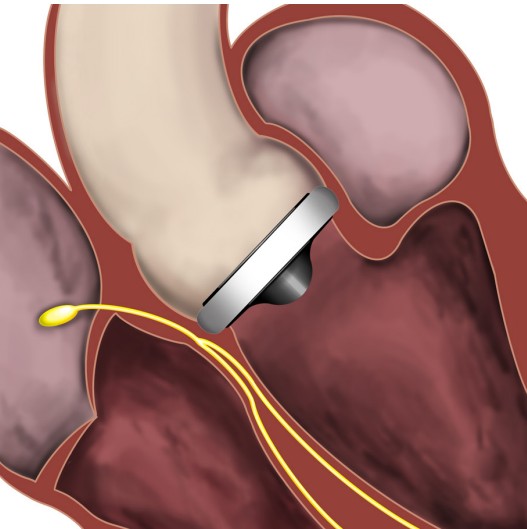

**Figure 2** The On-X prosthetic heart valve (On-X Life Technologies, Austin, Texas, USA; left panel) has an elongated subannular valve housing with a flared inlet that protrudes subannulary into the left ventricular outflow tract, in close proximity to the cardiac conduction system. This design distinguishes the On-X prosthetic valve from other contemporary bileaflet mechanical aortic valves, such as the St Jude Regent (St Jude Medical, St Paul, Minnesota, USA; right panel), that are implanted intra-/supraannulary and do not extend subannulary. Illustration: Magnus Dalén.

The potential risks associated with permanent pacemaker requirement are particularly relevant for younger patients since they have a longer life expectancy.[27]

The incidence of permanent pacemaker implantation in the current study (4.4%) is in line with previous studies of mechanical aortic valve replacement in younger adults, reporting rates of 2.5%–5%.[28 29] Few head-to-head comparisons of mechanical bileaflet prosthetic valves have been reported[30] and none of these have analysed permanent pacemaker implantation rates between different valves. Generally, there are very few reports regarding rates of permanent pacemaker implantation after mechanical aortic valve replacement and most prior studies concerning mechanical prosthetic valves are completely lacking information on permanent pacemaker rates.

Compression on the subannunlar tissue, and thereby subsequent possible compression of the cardiac conduction system can induce conduction disturbances necessitating permanent pacemaker implantation. Because bioprosthetic valves with an extended subannular component, such as transcatheter and rapid deployment prosthetic valves, have been associated with a markedly increased risk for permanent pacemaker implantation,[5–7] we aimed to analyse whether this would also be true for the On-X prosthetic valve. Despite its' elongated subannular valve housing, our results demonstrated that the On-X prosthetic heart valve did not confer a higher risk of permanent pacemaker implantation compared with other conventional intra-/supraannulary bileaflet mechanical valves. This might be explained by the differences in design and annular fixation between the On-X prosthetic valve and transcatheter/rapid deployment prosthetic valves. These valves, unlike the On-X, are balloon- or self-expandable bioprosthetic valves that rely on radial forces for deployment of a subannular frame for stabilisation in the aortic annulus and left ventricular outflow tract. These valves are therefore oversized in relation to the aortic annulus. This is in contrast with the On-X valve, which is sutured to the aortic annulus with no expansion of the subannular valve component. Correct implantation requires that the On-X subannular valve housing fit into the left ventricular outflow tract and oversizing is thereby not possible.

## Limitations

This analysis has limitations. First, the data did not include preoperative conduction disorders known to increase the risk for postoperative pacemaker requirement, such as bundle branch block, as well as other risk factors for postoperative pacemaker requirement such as mitral annular calcification, left ventricular outflow tract calcification and surgical suture technique that was used for valve implantation. We were therefore not able to adjust for such risk factors. We were unable to account for possible centre effects, owing to a low number of events (pacemaker implantations) in the On-X group. Second, we did not have information about specific indications of postoperative pacemaker implantation. Third, we included implantable cardiac defibrillators in the primary outcome measure, in order to capture patients who had dual indications for both pacing and defibrillation. It is possible that some of these patients only had indication for defibrillation.

## CONCLUSIONS

The On-X prosthetic heart valve was associated with a similarly low risk for permanent pacemaker implantation after aortic valve replacement compared with other conventional bileaflet mechanical valves. This indicates that the On-X elongated subannular valve housing does not seem to inflict on the cardiac conduction system.

**Acknowledgements** We thank the SWEDEHEART steering committee for providing data for this study.

**Contributors** All authors (MD, MP, NG, US) have made substantial contributions to the conception or design of the work; or the acquisition, analysis, or interpretation of data for the work; and drafting the work and revising it critically for important intellectual content; and made final approval of the version to be published; and agreed to be accountable for all aspects of the work in ensuring that questions related to the accuracy or integrity of any part of the work are appropriately investigated and resolved. MD is the guarantor.

**Funding** This work was supported by a donation from Mr. Fredrik Lundberg (to MD), the Swedish Heart-Lung Foundation (grant numbers 20180400 and 20190533 to US, and 20190570 to NG), Åke Wiberg Foundation (grant number M18-0016 to US), Region Stockholm (ALF project) (grant number 20180114 to US), the Swedish Society of Medicine (grant number SLS-934749 to NG), the Capio Research Foundation (grant number 2019-3239 to NG), the Eva and Oscar Ahrén Research Foundation (to NG), and the Seraphim Hospital Foundation (to NG).

**Competing interests** None declared.

**Patient consent for publication** Not applicable.

**Ethics approval** The study was approved by the Swedish Ethical Review Authority and the need for informed consent was waived (Registration number: 2019-04131).

**Provenance and peer review** Not commissioned; externally peer reviewed.

**Data availability statement** All data relevant to the study are included in the article or uploaded as supplementary information.

**ORCID iDs**
Magnus Dalén http://orcid.org/0000-0001-5352-5132
Natalie Glaser http://orcid.org/0000-0002-4386-1041
Ulrik Sartipy http://orcid.org/0000-0003-2707-0263

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
