## [Reviewer comments · BMJ Open]

ARTICLE DETAILS

TITLE (PROVISIONAL)	Permanent pacemaker implantation after On-X surgical aortic valve replacement: SWEDEHEART observational study
AUTHORS	Dalén, Magnus; Persson, Michael; Glaser, Natalie; Sartipy, Ulrik

VERSION 1 – REVIEW

REVIEWER	Ayman Elbadawi University of Texas
REVIEW RETURNED	02-Mar-2021

GENERAL COMMENTS	I read with interest the manuscript by Dr. Dalen et al. The authors report the results of their observational analysis using a national cohort in Sweden, to evaluate the rates of permanent pacemaker implantation (PPM) after On-X mechanical aortic valve replacement (AVR) versus other bileaflet mechanical leaflets. Their study period spanned 2005 to 2018. A total of 2,602 patients were identified and 581 patients received On-X valve and 2021 patients received a St Jude Masters/Regent (n=945) or Carbomedics Reduced valve (n=1076). After propensity score matching, there was no difference in the rate of PPM after On-X compared with other valves (3.6% vs. 4.0%). I would like to commend the authors on their work and well written manuscript. I have the following comments: 1-I would prefer to change the title to clearly indicate that the group of interest are patients undergoing surgical aortic valve replacement. 2-The variables used for confounding adjustment (mainly with propensity score matching) do not include hospital related variables. Given significant variability in proportion of On-x valves according to hospitals (hospital 8 had substantially higher proportion), it would be important to adjust for hospital stratification in the adjustment models. 3-Keeping with the prior comment, it would be interesting to explore any volume-outcome between rates of PPM and hospital/operators volume of mechanical valve replacement (particularly volume of On-X valve implantation). In other words, do hospitals/operators who perform more mechanical valves have lower rates of PPM post-operatively? 4-There are number of independent predictors for pacemaker implantation post AVR that are not captured in this analysis. Authors identify in the limitation that pre-existing conduction disorders were not captured, which is indeed a vital missing information. Additionally, it would be important to report data on mitral annular
---

	coalification and LVOT calcification, which have been both predictors of PPM after AVR. 5-Post-operative conduction disorders can present variably after surgeries, and might show up even months later. For example, some patients may develop post-procedure bifascicular or trifascicular blocks, and on follow up develop symptoms warranting pacemaker implantation. Hence, it is always preferable to estimate rates of PPM insertion at the longest available follow up period. The main outcome of the study was evaluation of PPM rate at “30-day”, while there appears to be data up to 90-days post-operative. Even though most PPM were within 30 days, still some patients got PPM beyond 30-days. It is unclear why authors chose the time cutoff at 30-days rather than 90-days. Please explain. 6-The main outcome of PPM implantation included those receiving implantable cardiac defibrillators. This is not entirely accurate (as hinted to by authors). In the current retrospective analysis, some patients might have received implantable cardiac defibrillators due to ventricular arrhythmia events post valve replacement, and not necessarily due to bradyarrhythmias or conduction system disorders (and authors state the indications for PPM were irretrievable in the current dataset). A separate analysis should be conducted to report the differential rates of permanent pacemaker implantation (not including defibrillators), this could be adopted as a sensitivity or secondary analysis, after adjustment for confounding factors (preferable propensity score matching). 7-As mentioned by authors the implantation rates increased during their study period. From 2005 to 2018, there has been substantial rise in the number of transcatheter aortic valve implantation (TAVI) procedures. It has been demonstrated in other studies, that the introduction of TAVI allowed better selection of lower risk patients to undergo surgical valve replacement, and expectedly, the rates of PPM should have plateaued if not dropped. Can authors comment on such rising trend in PPM implantation in their study. Are there any comparable data on the trend in patients comorbidities, demographics during the study years? 8-The discussion section could be improved by reporting comparative data in literature for rates and predictors of PPM after mechanical AVR.
--	---

REVIEWER	Stephen Froles Sunnybrook Health Sciences Centre
REVIEW RETURNED	07-Mar-2021

GENERAL COMMENTS	The manuscript attempts to determine in implantation of the Onyx valve is associated with a higher risk of permanent pacemaker implantation after mechanical AVR compared to other mechanical valves. The manuscript is very well written. The rationale is that the Onyx valve is by design, an intra-annular valve. That distinguishes it from other valves. The St Jude is partially intra-annular - the hinge mechanism is protected with an extension of the housing that extends intra-annular - this extension is aligned
---

	typically towards the non-left commissure and the septum in the middle of the right sinus, i.e. away from the membranous septum. The authors used SWEDEHEART registries as the data source for the study. The authors reported crude rates and also 3 sets of adjusted rates - PSMed, IPTW, and overlap weights. The balance after adjusted was very good for each of the PS adjustment methods according to SMDs. The authors reported results as the proportion of implants at 30 days. The results are consistent with the authors' conclusions that there is not increased risk of PPM seen with with Onyx valve. I have the following comments. There may be an era effect; the Onyx valves were implanted infrequently in the early years of the cohort but with increasing frequency in the later years. Did the authors include a variable for era in the LR model to derive the PSs? Or did they include a variable for era in the analyses using the PSs? There is great variability between institutions, with some institutions implanting no Onyx valves. Did the authors attempt to make adjustments for institution in their analyses in some manner? If not, this should be stated as a limitation. Do the authors have any results of the postoperative ECG? Do they know what proportion of the patients have a new LBBB? Are the KM curves crude or using the PS adjusted analyses? If the latter, the authors need to describe appropriate methods for survival analysis using matched or weighted data. Comparisons of the PS adjusted cohorts: For the PSMed cohorts, the authors should use McNemar rather than Chi-square, as the cohorts are matched rather than independent. A similar comment applies to the weighted analyses, but a statistician should say whether Chi-square is appropriate, or whether another test which accounts for the similarity of the cohorts in the 2 weighted analyses is appropriate. I do not think this will affect the authors' conclusions in any way. Minor Comments: Onyx is the experimental group - suggest that the columns for Onyx precede the columns for the controls in the tables. (The authors do provide results for the Onyx before the controls in the Abstract and Results section). Please include a figure which includes the mirror histograms of the PSs before and after matching. Please include height/weight/BSA to Table 1 and all other tables of baseline characteristics. Please consider adding type of hospital to Table 1 and all other tables of baseline characteristics - if all hospitals are urban/teaching hospitals in Sweden, then this comment does not apply.
--	---

	There are many suture methods for AVR, which may influence the results and is a limitation.
--	---

VERSION 1 – AUTHOR RESPONSE

Reviewer: 1

Dr. Ayman Elbadawi, University of Texas

Comments to the Author:

I read with interest the manuscript by Dr. Dalen et al. The authors report the results of their observational analysis using a national cohort in Sweden, to evaluate the rates of permanent pacemaker implantation (PPM) after On-X mechanical aortic valve replacement (AVR) versus other bileaflet mechanical leaflets. Their study period spanned 2005 to 2018. A total of 2,602 patients were identified and 581 patients received On-X valve and 2021 patients received a St Jude Masters/Regent (n=945) or Carbomedics Reduced valve (n=1076). After propensity score matching, there was no difference in the rate of PPM after On-X compared with other valves (3.6% vs. 4.0%).

I would like to commend the authors on their work and well written manuscript. I have the following comments:

1-I would prefer to change the title to clearly indicate that the group of interest are patients undergoing surgical aortic valve replacement.

Author's reply:

Thank you for taking the time to review our manuscript and for providing constructive criticism. As suggested by the reviewer, we have changed the title in the revised version of the manuscript and it now reads as follows:

“Permanent pacemaker implantation after On-X surgical aortic valve replacement: SWEDEHEART observational study”

2-The variables used for confounding adjustment (mainly with propensity score matching) do not include hospital related variables. Given significant variability in proportion of On-x valves according to hospitals (hospital 8 had substantially higher proportion), it would be important to adjust for hospital stratification in the adjustment models.

Author's reply:

The reviewer raises an important point: Is it possible that hospital/center influence the rate of pacemaker implantations differentially in On-X AVR vs. other AVR?

We considered the possibility of hospital/center effects during the planning and analyses in this study. However, owing to the relatively low number of permanent pacemaker implantations in On-X patients in each hospital (numbers varied from 0 to 12), it was not possible to reliably investigate possible center effects.

We provide more details below:

This question cannot be addressed in the 2 hospitals that did not use On-X valves at all, nor in the one hospital that only implanted 6 On-X valves (Supplemental Table 2 and Supplemental Figure 1). We performed a stratified analyses in the remaining 5 hospitals where the On-X implantation ratio varied from 7.6% to 74%. In 4 of the 5 hospitals, 1 to 3 patients with an On-X valve received a permanent pacemaker implantation. Because the number of pacemaker implantations was very low,

the results must be interpreted with caution. It is evident from the table below that data must be pooled and possible center effects could not be evaluated in this dataset.

The following text was added to the revised version of the manuscript in the Limitations section: “We were unable to account for possible center effects, owing to a low number of events (pacemaker implantations) in the On-X group.”

Number of pacemaker implants according to hospital and valve group.

Hospital	Control	On-X	p-value
1	22/413 (5.3%)	3/54 (5.6%)	0.944
2	22/227 (9.7%)	2/113 (1.8%)	0.007
3	17/304 (5.6%)	1/25 (4.0%)	0.737
6	9/243 (3.7%)	3/88 (3.4%)	0.899
8	2/101 (2.0%)	12/295 (4.1%)	0.327

Numbers are n (%).

Number of pacemaker implants according to hospital and valve group.

Hospital	Control	On-X	p-value
1	22/413 (5.3%)	3/54 (5.6%)	0.944
2	22/227 (9.7%)	2/113 (1.8%)	0.007
3	17/304 (5.6%)	1/25 (4.0%)	0.737
6	9/243 (3.7%)	3/88 (3.4%)	0.899
8	2/101 (2.0%)	12/295 (4.1%)	0.327

Numbers are n (%).

3-Keeping with the prior comment, it would be interesting to explore any volume-outcome between rates of PPM and hospital/operators volume of mechanical valve replacement (particularly volume of On-X valve implantation). In other words, do hospitals/operators who perform more mechanical valves have lower rates of PPM post-operatively?

Author's reply:

This is an interesting research question, but unfortunately, the association between hospital/operator volume and permanent pacemaker implantation rate cannot be reliably estimated within the current dataset. We do not have information identifying the operator/surgeon.

In the current study, the aim was to investigate if AVR with an On-X valve was associated with an increased risk of permanent pacemaker implantation. The question if there is an association between volume of mechanical AVR and subsequent need for pacemakers is beyond the scope of the current study. We believe that this question should be investigated separately in a study that is designed for that specific purpose.

4-There are number of independent predictors for pacemaker implantation post AVR that are not captured in this analysis. Authors identify in the limitation that pre-existing conduction disorders were not captured, which is indeed a vital missing information. Additionally, it would be important to report data on mitral annular coalification and LVOT calcification, which have been both predictors of PPM after AVR.

Author's reply:

Unfortunately, the Swedish national health-data registers that was used in our study do not contain information on mitral annular calcification or LVOT calcification. To acknowledge these limitations, the following text was added to the revised version of the manuscript:

"The data did not include preoperative conduction disorders known to increase the risk for postoperative pacemaker requirement, such as bundle branch block, mitral annular calcification or left ventricular outflow tract calcification."

5-Post-operative conduction disorders can present variably after surgeries, and might show up even months later. For example, some patients may develop post-procedure bifascicular or trifascicular blocks, and on follow up develop symptoms warranting pacemaker implantation. Hence, it is always preferable to estimate rates of PPM insertion at the longest available follow up period. The main outcome of the study was evaluation of PPM rate at "30-day", while there appears to be data up to 90-days post-operative. Even though most PPM were within 30 days, still some patients got PPM beyond 30-days. It is unclear why authors chose the time cutoff at 30-days rather than 90-days. Please explain.

Author's reply:

We decided on the time cut-off at 30 days because this is the most common definition in previous literature. We believe that this decision was reasonable because our data (Figure 1) showed that very few patients received a pacemaker between day 30 and 60, and no patients received a pacemaker from day 60 to day 120.

6-The main outcome of PPM implantation included those receiving implantable cardiac defibrillators. This is not entirely accurate (as hinted to by authors). In the current retrospective analysis, some patients might have received implantable cardiac defibrillators due to ventricular arrhythmia events post valve replacement, and not necessarily due to bradyarrhythmias or conduction system disorders (and authors state the indications for PPM were irretrievable in the current dataset). A separate analysis should be conducted to report the differential rates of permanent pacemaker implantation (not including defibrillators), this could be adopted as a sensitivity or secondary analysis, after adjustment for confounding factors (preferable propensity score matching).

Author's reply:

There were very few ICDs implanted within 30 days of AVR in our study: 1/21 (4.8%) in the On-X group and 5/94 (5.3%) in the group who received other mechanical valves. We performed a repeated

analysis, excluding patients who had an ICD, and found practically identical results as in our main analysis.

7-As mentioned by authors the implantation rates increased during their study period. From 2005 to 2018, there has been substantial rise in the number of transcatheter aortic valve implantation (TAVI) procedures. It has been demonstrated in other studies, that the introduction of TAVI allowed better selection of lower risk patients to undergo surgical valve replacement, and expectedly, the rates of PPM should have plateaued if not dropped. Can authors comment on such rising trend in PPM implantation in their study. Are there any comparable data on the trend in patients comorbidities, demographics during the study years?

Author's reply:

As the Reviewer correctly points out, the increased use of TAVI has allowed for a selection of the patients with highest surgical risk to undergo TAVI. Indeed, risk factors for permanent pacemaker implantation after surgical AVR include chronic comorbidities. However, more important risk factors are preoperative conduction disturbances and history of arrhythmia. Consequently, there are no reports that the increased use of TAVI should have decreased the risk for permanent pacemaker implantation after surgical AVR. We hypothesize that the slightly increased rate of permanent pacemaker implantation in Sweden during the current study period might be owing to e.g. increased use of continuous ECG monitoring during the complete hospital stay.

8-The discussion section could be improved by reporting comparative data in literature for rates and predictors of PPM after mechanical AVR.

Author's reply:

We agree with the Reviewer that comparative data regarding rates and predictors of permanent pacemaker implantation after mechanical AVR is of importance. In the Discussion section, we have stated that:

"The incidence of permanent pacemaker implantation in the current study (4.4%) is in line with previous studies of mechanical aortic valve replacement in younger adults, reporting rates of 2.5-5%.^{28, 29} Few head-to-head comparisons of mechanical bileaflet prosthetic valves have been reported³⁰ and none of these have analyzed permanent pacemaker implantation rates between different valves."

However, as we also state in the Discussion section, there are very few reports regarding rates of and risk factors for of permanent pacemaker implantation after mechanical aortic valve replacement in the literature. The cited references (reference 28, 29, 30) are the most reliable references we have been able to find in the previous literature.

Reviewer: 2

Dr. Stephen Fremes, Sunnybrook Health Sciences Centre

Comments to the Author:

The manuscript attempts to determine if implantation of the Onyx valve is associated with a higher risk of permanent pacemaker implantation after mechanical AVR compared to other mechanical valves.

The manuscript is very well written.

The rationale is that the Onyx valve is by design, an intra-annular valve. That distinguishes it from other valves. The St Jude is partially intra-annular - the hinge mechanism is protected with an extension of the housing that extends intra-annular - this extension is aligned typically towards the non-left commissure and the septum in the middle of the right sinus, i.e. away from the membranous septum.

The authors used SWEDEHEART registries as the data source for the study.

The authors reported crude rates and also 3 sets of adjusted rates - PSMed, IPTW, and overlap weights. The balance after adjusted was very good for each of the PS adjustment methods according to SMDs. The authors reported results as the proportion of implants at 30 days.

The results are consistent with the authors' conclusions that there is not increased risk of PPM seen with with Onyx valve.

I have the following comments.

1. There may be an era effect; the Onyx valves were implanted infrequently in the early years of the cohort but with increasing frequency in the later years. Did the authors include a variable for era in the LR model to derive the PSs? Or did they include a variable for era in the analyses using the PSs?

Author's reply:

Thank you for the constructive and positive feedback.

We did not include a variable for era in the logistic regression model for derivation of propensity scores, and we did not include a variable for era in the outcome analyses. Intuitively, it is not obvious (but not impossible) that the risk for permanent pacemaker implantation would change differentially with time in On-X AVR vs. other valve AVR.

As shown in Supplemental Table 1 and Supplemental Figure 1, from 2005 to 2008, the proportion of On-X valves increased from 3% to 18%. From 2009 to 2018, the proportion of On-X valves was fairly stable and varied between 24% to 37%.

The total number of yearly pacemaker implantations in the total study population was low and varied from 3 (2.3%) to 15 (6.8%). In the On-X group, the yearly total number of pacemaker implantations was very low and varied between 0 to 6 (8.2%). Owing to the low number of yearly pacemaker implantations (particularly in the On-X group) it was not possible to reliably estimate or account for a possible era effect in this dataset.

Number of pacemaker implants according to era and valve group.

Year of surgery	Control	On-X	p-value
2005–2010	43/1067 (4.0%)	6/168 (3.6%)	0.777
2011–2018	51/954 (5.3%)	15/413 (3.6%)	0.175

Numbers are n (%).

2. There is great variability between institutions, with some institutions implanting no Onyx valves. Did the authors attempt to make adjustments for institution in their analyses in some manner? If not, this should be stated as a limitation.

Author's reply:

Please see our response to Reviewer #1, comment #2.

The following text was added to the revised version of the manuscript in the Limitations section: "We were unable to account for possible center effects, owing to a low number of events (pacemaker implantations) in the On-X group."

3. Do the authors have any results of the postoperative ECG? Do they know what proportion of the patients have a new LBBB?

Author's reply:

Unfortunately, the Swedish national health-data registers that were used in this study do not contain information regarding postoperative ECG. We acknowledge the lack of information regarding conduction disturbances in the Study Limitations section.

4. Are the KM curves crude or using the PS adjusted analyses? If the latter, the authors need to describe appropriate methods for survival analysis using matched or weighted data.

Author's reply:

As stated in the Figure legend (Figure 1), the graph shows the Kaplan-Meier estimated permanent pacemaker implantation rate in propensity score matched patients who received an On-X valve compared to other valves. However, we did not use survival analysis: the Kaplan-Meier plots were only used to give a visual overview of which postoperative day the pacemaker implantation occurred. No results (event rates) or statistical inferences were based on the Kaplan-Meier analyses, they are simply used for a visual representation of the data.

5. Comparisons of the PS adjusted cohorts: For the PS Med cohorts, the authors should use McNemar rather than Chi-square, as the cohorts are matched rather than independent. A similar comment applies to the weighted analyses, but a statistician should say whether Chi-square is appropriate, or whether another test which accounts for the similarity of the cohorts in the 2 weighted analyses is appropriate. I do not think this will affect the authors' conclusions in any way.

Author's reply:

As suggested by the reviewer, we replaced the Chi-square test in the propensity score matched analysis with McNemar's test. In the analyses in the weighted populations, it is appropriate to use a standard Chi-square test.

The manuscript was revised to reflect this change.

Statistical methods

"In the matched and weighed populations, the percentage of pacemaker implants and 95% confidence interval was compared between the groups using McNemar's and chi-square tests, respectively."

Minor Comments:

6. Onyx is the experimental group - suggest that the columns for Onyx precede the columns for the controls in the tables. (The authors do provide results for the Onyx before the controls in the Abstract and Results section).

Author's reply:

We would prefer to keep the column order because it seems to us that this is common place for observational studies (as opposed to RCTs) in the BMJ, but are of course prepared to change on the request of the Editors.

7. Please include a figure which includes the mirror histograms of the PSs before and after matching.

Author's reply:

As suggested by the Reviewer, we have added a figure that shows the distribution of propensity scores before and after matching to the Supplement.

Supplemental Figure 3 was added to the revised Supplemental Material.

Supplemental Figure 3. Histogram of propensity scores before and after matching.

8. Please include height/weight/BSA to Table 1 and all other tables of baseline characteristics.

Author's reply:

We have added a new Supplemental Table including height, weight and body surface area. We believe that it would be less than ideal to include these variables in Table 1 because they were not included in the propensity score model. Therefore, we fear that including them in Table 1 could cause confusion and instead we provide these data in a separate table.

Supplemental Table 7 was added to the revised Supplemental Material.

9. Please consider adding type of hospital to Table 1 and all other tables of baseline characteristics - if all hospitals are urban/teaching hospitals in Sweden, then this comment does not apply.

Author's reply:

All 8 hospitals performing cardiac surgery in Sweden can be considered urban/teaching hospitals. Transparent and open reporting of volumes and quality of care are available since many years through the Swedeheart webpages, and clinical outcomes are similar between centers.

10. There are many suture methods for AVR, which may influence the results and is a limitation.

Author's reply:

As suggested by the Reviewer, we have added the following text to the Study Limitations section in the revised version of the manuscript:

“The data did not include preoperative conduction disorders known to increase the risk for postoperative pacemaker requirement, such as bundle branch block, as well as other risk factors for postoperative pacemaker requirement such as mitral annular calcification, left ventricular outflow tract calcification, and surgical suture technique that was used for valve implantation.

VERSION 2 – REVIEW

REVIEWER	Stephen Fremes Sunnybrook Health Sciences Centre
REVIEW RETURNED	13-Jul-2021
GENERAL COMMENTS	The original manuscript was very well written. The authors responded well to all earlier comments. No further comments. Suggest accept as is.